# Characterization of Patients with Pulmonary Arterial Hypertension: Data from the Polish Registry of Pulmonary Hypertension (BNP-PL)

**DOI:** 10.3390/jcm9010173

**Published:** 2020-01-08

**Authors:** Grzegorz Kopeć, Marcin Kurzyna, Ewa Mroczek, Łukasz Chrzanowski, Tatiana Mularek-Kubzdela, Ilona Skoczylas, Beata Kuśmierczyk, Piotr Pruszczyk, Piotr Błaszczak, Ewa Lewicka, Danuta Karasek, Katarzyna Mizia-Stec, Michał Tomaszewski, Wojciech Jacheć, Katarzyna Ptaszyńska-Kopczyńska, Małgorzata Peregud-Pogorzelska, Anna Doboszyńska, Agnieszka Pawlak, Zbigniew Gąsior, Wiesława Zabłocka, Robert Ryczek, Katarzyna Widejko-Pietkiewicz, Marcin Waligóra, Szymon Darocha, Michał Furdal, Michał Ciurzyński, Jarosław D. Kasprzak, Marek Grabka, Karol Kamiński, Piotr Hoffman, Piotr Podolec, Adam Torbicki

**Affiliations:** 1Department of Cardiac and Vascular Diseases, Jagiellonian University Medical College, John Paul II Hospital in Krakow, Ul. Prądnicka 80, 31-202 Krakow, Poland; 2Department of Pulmonary Hypertension, Thromboembolic Diseases and Cardiology, Centre of Postgraduate Medical Education, 05-400 Otwock, Poland; marcin.kurzyna@ecz-otwock.pl (M.K.);; 3Department of Cardiology Provincial Specialist Hospital Research and Development Center, 51-124 Wrocław, Poland; mroczeke@wp.pl (E.M.);; 4Cardiology Department, Medical University of Lodz, 91-347 Lodz, Poland; chrzanowski@ptkardio.pl (Ł.C.);; 5Department of Cardiology, Poznan University of Medical Sciences, 61-701 Poznan, Poland; tatianamularek@wp.pl; 63rd Department of Cardiology, Faculty of Medical Sciences in Zabrze, Medical University of Silesia, 41-800 Katowice, Poland; iskoczylas@tlen.pl; 7Department of Congenital Heart Disease Institute of Cardiology, 04-628 Warsaw, Poland; bkusmier@gmail.com (B.K.);; 8Department of Internal Medicine and Cardiology, Medical University of Warsaw, 02-005 Warsaw, Poland; piotr.pruszczyk@wum.edu.pl (P.P.);; 9Department of Cardiology, Cardinal Wyszynski Hospital, 20-718 Lublin, Poland; blaszcz12345@interia.pl; 10Department of Cardiology and Electrotherapy Medical University of Gdansk, 80-211 Gdansk, Poland; elew@gumed.edu.pl; 112nd Department of Cardiology, Faculty of Health Sciences, Collegium Medicum, Nicolaus Copernicus University, 85-168 Bydgoszcz, Poland; danuta.karasek@op.pl; 12I Katedra i Klinika Kardiologii, Wydział Lekarski w Katowicach, Śląski Uniwersytety Medyczny w Katowicach, 40-635 Katowice, Poland; kmiziastec@gmail.com (K.M.-S.); marekgrabka@interia.pl (M.G.); 13Department of Cardiology, Medical University of Lublin, 20-090 Lublin, Poland; mdtomaszewski@wp.pl; 142nd Department of Cardiology, School of Medicine with Dentistry Division in Zabrze, Medical University of Silesia in Katowice, 41-800 Zabrze, Poland; wjachec@interia.pl; 15Department of Cardiology, Medical University of Bialystok, 15-276 Bialystok, Poland; kasia.ptaszynska@op.pl; 16Department of Cardiology, Pomeranian Medical University, Szczecin, 70-111 Szczecin, Poland; m1peregud@gmail.com; 17Pulmonary Department, University of Warmia and Mazury, 10-357 Olsztyn, Poland; anna.doboszynska@wp.pl; 18Department of Invasive Cardiology, Polish Academy of Sciences, Mossakowski Medical Research Centre, Central Clinical Hospital of the Ministry of Interior, 02-507 Warsaw, Poland; a.pawlak1@wp.pl; 19Department of Cardiology, School of Health Sciences, Medical University of Cardiology in Katowice, 40-635 Katowice, Poland; zbgasiar@gmail.com; 20Oddział Kardiologii i Kardiologii Inwazyjnej, Oddział Intensywnego Nadzoru Kardiologicznego, Samodzielny Publiczny Wojewódzki Szpital Zespolony w Szczecinie, 71-455 Szczecin, Poland; wieslawa.zablocka@wp.pl; 21Department of Cardiology and Internal Medicine, Military Institute of Medicine in Warsaw, 04-141 Warsaw, Poland; raryczek@gmail.com; 22Department of Cardiology, Copper Health Center, 59-300 Lubin, Poland; kwidejko@wp.pl; 23Department of Population Medicine and Civilization Diseases Prevention, Medical University of Bialystok, 15-269 Bialystok, Poland; fizklin@wp.pl

**Keywords:** pulmonary arterial hypertension, registry, epidemiology, prevalence, incidence

## Abstract

Current knowledge of pulmonary arterial hypertension (PAH) epidemiology is based mainly on data from Western populations, and therefore we aimed to characterize a large group of Caucasian PAH adults of Central-Eastern European origin. We analyzed data of incident and prevalent PAH adults enrolled in a prospective national registry involving all Polish PAH centers. The estimated prevalence and annual incidence of PAH were 30.8/mln adults and 5.2/mln adults, respectively and they were the highest in females ≥65 years old. The most frequent type of PAH was idiopathic (*n* = 444; 46%) followed by PAH associated with congenital heart diseases (CHD-PAH, *n* = 356; 36.7%), and PAH associated with connective tissue disease (CTD-PAH, *n* = 132; 13.6%). At enrollment, most incident cases (71.9%) were at intermediate mortality risk and the prevalent cases had most of their risk factors in the intermediate or high risk range. The use of triple combination therapy was rare (4.7%). A high prevalence of PAH among older population confirms the changing demographics of PAH found in the Western countries. In contrast, we found: a female predominance across all age groups, a high proportion of patients with CHD-PAH as compared to patients with CTD-PAH and a low use of triple combination therapy.

## 1. Introduction

Pulmonary arterial hypertension (PAH) is a rare disease in which progressive narrowing of the pulmonary arterial lumen leads to a rise in pulmonary arterial pressure and pulmonary vascular resistance (PVR) [1]. Currently, most of our knowledge of PAH comes from registries originating in Western populations [2,3,4,5,6,7,8]. However, taking a global view of PAH epidemiology shows important geographical differences in the characteristics of PAH patients [9,10,11].

In Poland, as in other countries of the former Eastern Bloc, the transformation of the political system, started in the early 1990s, opened the health care system to influences from Western medicine. Still, it was not until 2008, that the first PAH targeted therapies were formally reimbursed. Consequently, advanced diagnostics and treatments of PAH were available about two decades later than in the Western countries.

The Data Base of Pulmonary Hypertension in the Polish Population (Baza Nadciśnienia Płucnego; BNP-PL, https://clinicaltrials.gov/ct2/show/NCT03959748) is the first multicenter prospective registry of adult and pediatric PAH and chronic thromboembolic pulmonary hypertension (CTEPH) patients created in any Central-Eastern European country [12].

In the present report, we present the prevalence of PAH in Polish adults and their baseline characteristics including hemodynamics, exercise capacity, comorbidities and disease management. 

## 2. Experimental Section

### 2.1. Design of the BNP-PL Registry—PAH Adults Arm

The design of the BNP-PL registry, enrollment criteria and data collection were recently described in detail [12]. The PAH adult arm of the registry enrolls adults with PAH from all 21 PH reference centers in Poland accredited by the National Health Fund to treat PAH. The protocol of the study was reviewed and accepted by the Bioethical Committee of Physicians and Dentists Chamber in Krakow. 

### 2.2. Analysis of PAH Patients

For the purpose of the present study, we analyzed the data of newly and previously diagnosed PAH adult (≥18 years old) patients who were under the care of the participating centers between 1 March 2018 and 30 September 2018. The diagnostic algorithm was based on current recommendations [11,13,14,15,16,17,18,19]. Newly diagnosed patients were those whose diagnosis was established since 1 March 2018 (termed ‘incident cases’). Patients who were diagnosed earlier have been classified as ‘prevalent cases’. PAH was defined and classified according to the guidelines of the European Society of Cardiology [15] (ESC; Methods 1 (in Appendix A)). The main goal of our registry is to describe current practice and outcomes in patients with PAH therefore the registry is entirely observational and the protocol of the study does not require any additional patient visits or diagnostic tests, and does not influence the management of patients. We also did not prepare any specific protocol for genetic testing. The diagnosis of IPAH was at the discretion of coinvestigators who managed the enrolled patients in their centers. However, the diagnosis must have fulfilled the criteria recommended by the European Society of Cardiology. This means that patients were diagnosed for PAH when left heart disease, chronic thromboembolic disease, and pulmonary diseases were excluded. IPAH could have been diagnosed only in patients in whom congenital heart defects, connective tissue disease, HIV infection, portal hypertension and possible relation to drug use were excluded. We enrolled in our study reference centers of PAH accredited by the National Health Fund which ensures validity and credibility of the diagnostic algorithms. However, we cannot exclude some variability in the diagnostic approach of different centers which we did not study in detail for the purpose of the present analysis.

As genetic testing is not routinely performed in PAH patients in Poland currently, we cannot make conclusions regarding the prevalence of heritable PAH (HPAH) from our registry. Accordingly, IPAH patients and patients in whom HPAH was diagnosed have been considered as a single group IPAH/HPAH. As PAH treatment can only be reimbursed in accredited centers, we believe that almost all patients diagnosed with PAH have been included in the project. 

### 2.3. Risk Assessment

We calculated the initial risk of all incident patients based on the recently proposed grading system of the Swedish PAH Register (SPAHR; Methods 2 (in Appendix A)) [6,20]. The risk of prevalent patients was illustrated by the proportion of patients who achieved a low, intermediate or high risk range for different risk determinants including World Health Organization Functional Class (WHO FC), 6-min walking distance (6 MWD), N-terminal pro-brain natriuretic peptide (NT-proBNP) or BNP, right atrial area (RAA), right atrial pressure (RAP), and cardiac index (CI). 

### 2.4. Statistical Analysis

Continuous variables were reported as means (SD). Categorical variables were reported as counts and percentages. Continuous variables were checked for normal distribution with chi-square test. For the comparison of continuous variables between 2 groups, we used the Student’s *t* test and for categorical variables, the χ^2^ test with Yate’s correction as needed. The significance level was set at alpha level of 0.05. Statistical analysis was performed with the use of Dell Inc. (2016), Dell Statistica (data analysis software system), (version 13, Dell, Texas, TX, USA) software.dell.com.

## 3. Results

### 3.1. Study Group 

A total of 1002 adult patients of Caucasian origin with PAH were entered into the electronic data base. The information pertaining to 32 of the patients was identified as being duplicated in the dataset and so these duplicate entries were discarded. Finally, 970 PAH patients met the study entry criteria based on hemodynamic criteria and these patients were enrolled in the present analysis. In a group of 96 (9.9%) incident cases the time from the first symptoms to first medical contact was 9.6 ± 27.3 months and from the first symptoms to PAH diagnosis was 12.6 ± 28.7 months. In 874 (90.1%) prevalent patients the time from PAH diagnosis to BNP-PL enrollment was 88.3 ± 104.9 months. Women accounted for the majority of the patients (*n* = 677; 69.8%). The mean age of patients was 46.8 ± 22.3 years (females 47.9 ± 22.1 vs. males 44.3 ± 22.7, *p* = 0.005) at diagnosis and 53.8 ± 17.9 years (females 55.0 ± 17.7 vs. males 50.9 ± 17.9; *p* < 0.0001) at enrollment. A significant number of patients (*n* = 322;33.2%) were at least 65 years old. Women accounted for the majority of patients in both the older (≥65 years) group (*n* = 239; 74.2%) and the younger (<65 years) group (438; 67.6%). Characterization of the study group is presented in Table 1. 

### 3.2. Prevalence, Incidence and Geographic Distribution of PAH 

The mean prevalence of PAH was 30.8 per million adults and the estimated incidence rate was 5.2 per million adults per year. They were the highest in older females (Figure 1). As shown in Figure A1 and Figure A2 (in Appendix B), the geographical distribution of PAH was heterogeneous, ranging from 14.4 per million to 46.6 per million adults in different regions. 

### 3.3. PAH Ssubgroups

The distribution of PAH subgroups according to the ESC classification at enrollment is shown in Figure 2 and Table 1 and Table 2. In the idiopathic/heritable PAH (IPAH/HPAH) group, 75 (16.9%) patients tested positive for acute vasoreactivity at diagnosis but at enrollment only 39 (8.7%) were still considered vasoreactive. In 13 patients, HPAH was confirmed by genetic testing. Importantly, genetic testing was not routinely performed in Polish PH centers.

### 3.4. Demographics and Prognostic Factors

At diagnosis, most patients (*n* = 699; 72.1%) were in WHO FC III, while at enrollment in FC II (*n* = 455; 46.9%). Levels of the main prognostic factors of the study group are shown in Table 1. In incident cases the baseline risk score was 2.0 ± 0.4. Most incident patients were in the intermediate risk (*n* = 69; 71,9%), followed by 14 (14,6%) in the high and 13 (13,5%) in the low risk. As shown in Table 3 most prevalent patients had their prognostic risk factors of PAH mortality in the range of intermediate or high risk. Most of them were able to achieve normalization of resting CI and RAP but only a quarter of them were able to normalize the RAA.

### 3.5. Comorbidities and PAH Specific Medications at Enrollment 

The PAH-specific treatments at the time of enrollment among study patients are shown in Table 4. All patients received PAH targeted therapies, apart from 39 patients with reactive IPAH who only received calcium channel blockers for their condition, and 4 patients who did not receive any PAH specific medication (who either did not meet the inclusion criteria for PAH specific reimbursement or did not agree to the proposed treatment). The numbers of patients treated with monotherapy and combination is shown in Table 4. In Figure A3 (in Appendix B), we show the use of different PAH targeted therapies. In Table 5, we show comorbidities and non-PAH specific treatments identified in PAH patients at enrollment.

### 3.6. Prevalent and Incident Cases

The incident as compared to the prevalent cases (Table 6) were older, had more comorbid conditions, and higher proportion of PAH associated with connective tissue disease (CTD-PAH) to PAH associated with congenital heart disease (CHD-PAH).

## 4. Discussion

The present analysis of the BNP-PL registry shows demographics, treatment and the burden of coexisting diseases in a large group of Caucasian adults of Central-Eastern European origin with all types of PAH. 

Both, the prevalence and incidence of PAH were the highest among older population which confirms the changing demographics of PAH found previously in the Western countries. In contrast to the Western European and US registries we found a female predominance not only in the younger patients but also in an older population, a high proportion of patients with CHD-PAH as compared to patients with CTD-PAH, and the low use of triple combination therapy.

### 4.1. Prevalence of PAH

PAH has been previously shown to be a rare disease with a prevalence ranging from 15 to 52 cases per million population. In the Scottish study [21] based on data from the national reference center responsible for the diagnosis and treatment of all PAH cases in Scotland, PAH was identified in 26 patients per million population. In the French PAH registry [22], based on data from 17 University hospitals, the prevalence of PAH was lower than in the Scottish study and was estimated at 15 cases per million adults. The authors reported this data to be an underestimate since a significant number of patients were known to be treated outside the reference centers which were included in the registry. Additionally, they reported marked differences in regional prevalence of PAH, ranging from five to 25 cases per million population, indicating that many patients were still unidentified or not referred to specialized centers. Our estimations of PAH prevalence are close to 30 patients per million adult population. This number refers to all PAH patients who had been diagnosed with PAH in centers who have a contract with the National Health Fund (NHF) for treatment for PAH. As NHF is the only health payer in Poland, we assume that almost all Polish patients diagnosed with PAH were enrolled in our registry. Nevertheless, we believe that our data still represents a lower estimate for PAH prevalence and that the level of awareness of this disease in some regions is insufficient. We base this assumption on the marked differences in regional prevalence of PAH.

### 4.2. Age and Sex Distribution

The prevalence and incidence rate of PAH in our study were the highest in older females. Of interest was also the finding that the incident patients were about 10 years older than prevalent ones. The rise in the mean age of PAH patients has been previously observed in other studies. In the US National Institutes of Health (NIH) registry conducted in the 1980s [23], the mean age of enrolled patients with primary pulmonary hypertension (currently IPAH) was 36.4 years while in the REVEAL study, recruiting patients 30 years later, the mean age in the IPAH group was 53 years [4,24]. In another study [25] the mean age of systematically observed incident IPAH patients increased from 45 years in 2001–2003 to 52 years in 2007–2009. This demographic shift was also reflected in the European COMPERA registry [26] in which incident PAH patients were in their 60 s at the time of diagnosis. These changing demographics for PAH patients has usually been accounted for by the aging of the populations of Western countries, and increasing awareness of PH among physicians and patients.

Our data confirm the predominance of female patients across major types of PAH. We found a female-to-male ratio of 2.3 in the total group of PAH patients and 2.5 in the group of patients with IPAH/HPAH, figures similar to those found in most other PAH registries in the Western Europe. In the French and Swedish registry, this ratio was 1.9 and in the Portuguese registry 2.4 [27]. Interestingly, the female predominance was significantly lower in PAH patients enrolled to the COMPERA study, in which the female-to-male ratio was 1.5 [26]. This proportion was related to the age at PAH diagnosis with the female-to-male ratio of 2.3 in patients aged 18–65 years, and 1.2 among the older patients. Similar results where shown in the Swedish registry, where the female-to-male ratio ranged from 1.9 in patients aged less than 45 years to 0.75 in patients aged at least 75 years [6]. In contrast, our study found a female predominance across all age groups. We hypothesize that it resulted from the fact that a significant proportion of PAH males did not achieve an older age due to increased mortality from PAH or other diseases as compared to females. Importantly, Polish males live 7.8 years shorter than Polish females. This difference is much lower in the German and Swedish population in which the life expectancy of males as compared to females is respectively 4.6 years and 3.5 years lower [https://www.who.int/gho/countries/en/]. Moreover, a high prevalence of cardiac and pulmonary diseases in elderly males in Poland may prohibit the proper diagnosis of PAH.

### 4.3. Types of PAH

The most frequent type of PAH in the BNP-PL registry was IPAH followed by CHD-PAH and CTD-PAH. The high proportion of CHD-PAH (36.7%) to CTD-PAH (13.6%) patients marks out our database as different from those of other Western European (11.3% vs. 15.3% in the French) or US registries (11.5% vs. 49.9% in the US REVEAL) [28,29]. In the Portuguese registry, the proportions of CHD-PAH and CTD-PAH were equal, while a higher ratio of CHD-PAH patients was found in the Chinese registry [30]. In the latter, CHD-PAH patients comprised 43% of the whole PAH group, being the most frequent type of PAH. The high prevalence of CHD-PAH in China has been attributed to limited access to corrective cardiac surgery in the past which can also explain our findings. The predominance of CHD-PAH found in prevalent cases is no longer seen among newly diagnosed patients which may result both from decreasing incidence of CHD-PAH and higher survival rates in CHD-PAH as compared to CTD-PAH subjects. In the PAH-CHD group, most patients presented with Eisenmenger’s syndrome developed as a complication of ventricular septal defects or complex heart defects. Similar proportions were shown previously by other groups [28].

Still the relatively low prevalence of CTD-PAH is noteworthy. Based on the prevalence of systemic sclerosis (SSc) in Poland (9.4 per 100,000) [31] we estimated the prevalence of PAH at 1.8 % in the SSc group, which is significantly lower than that of other European cohorts. For example, in a large population of SSc patients from France and Italy [32], PAH prevalence was 3.6%. We consider our data as a signal that CTD-PAH patients are lost, either by never getting a diagnosis or by not reaching the designated PAH centre.

### 4.4. Therapy, Comorbid Conditions and Risk

According to the current recommendations most PAH patients should receive a combination of PAH specific therapies. However, given the high costs of specific PAH treatment, the use of particular types of therapy may be a direct effect of the reimbursement policy and healthcare organization. 

In the BNP-PL registry, more than a half of patients were treated with a combination therapy. This data parallels a recent analysis of other European registries. In COMPERA, combination therapy was used in 40.3%, 49.6% and 57.5% of patients in 1-, 2-, and 3-year follow-ups respectively while in the French registry dual or triple combination therapy was initiated in 43% of patients within three months after PAH diagnosis. The low use of triple combination in our study may be as a result of the fact that it was not reimbursed in Poland until November 2018.

Prostacyclin analogs (treprostinil, epoprostenol, or inhaled iloprost) were used by 28.4% of the total group and by 38.5% of IPAH patients. Of note is the relatively high use of prostacyclin in patients who were in WHO FC IV in our study (65.8%) when compared to the REVEAL registry (47.6%) [33]. 

We found that the proportion of PAH patients who were using home oxygen therapy in our study was 10.6% which is low when compared to data from other registries including the US REVAL registry (40.3%) [4], the SPAHR registry (33%) [34] and the Spanish registry of patients treated with inhaled iloprost (38%) [35]. This may result from several factors including the lack of data from randomized trials to support oxygen supplementation in PAH subjects, quite liberal recommendations of the ESC for oxygen use in PAH (use only in patients with blood O_2_ pressure consistently >60 mmHg), and a high proportion of patients with CHD-PAH in our study. According to the ESC guidelines in the latter subpopulation of PAH patients the use of supplemental O_2_ therapy should be considered only in cases in which it produces a consistent increase in arterial O_2_ saturation and reduces symptoms. We can not also exclude the role of some difficulties with access to oxygen generators.

The baseline risk of our incident PAH patients calculated based on the ESC risk predictors was similar to other major PAH registries. It is however important to note that other factors not included in our analysis may also impact individual patient’s risk [36].

The prevalence of coronary artery disease and its classic risk factors, such as hypertension and diabetes was relatively high in patients with IPAH and CTD-PAH, which would suggest that left ventricular dysfunction contributes to the development of pulmonary hypertension in some patients [37]. However, we enrolled into our study only patients with PAWP ≤ 15 mmHg which establishes the diagnosis of precapillary pulmonary hypertension. Additionally, our data is in line with previous studies [25] which show the changing demographics of PAH. It is interesting to note the high usage of therapies generally not advised in PAH such as anticoagulants, beta blockers, and angiotensin enzyme inhibitors. This, however, seems to be due to comorbid conditions such as atrial fibrillation or flutter, ischemic heart disease, and myocardial infarction in the patients’ history. 

Recently, targeting the negative consequences of chronic sympathetic nerve activation with use of beta-blockers, and renal or pulmonary artery denervation has attracted interest. Till now, however, no clear demonstration of a favorable benefit-to-risk ratio of these therapies in PAH patients has been provided and multicenter randomized trials are needed to adapt their use in clinical practice [38].

In our registry we found that a significant number of patients (*n* = 127, 13%) had a diagnosis of COPD or asthma. Similar proportions (21.9%) were shown in the largest PAH registry conducted in the United States (REVEAL registry) [4]. This means that despite suffering from obstructive lung diseases some patients had been diagnosed with group 1 pulmonary hypertension (PAH). In that way COPD or asthma in a mild form were considered as a comorbid condition in a PAH patient rather than as an etiology of pulmonary hypertension. This approach has been recently supported by the experts of the 6th World Symposium on Pulmonary Hypertension [39], who suggest that in patients who have pulmonary hypertension and obstructive lung disease with FEV1 >60% and minimal parenchymal changes, the primary diagnosis should be PAH rather than group 3 pulmonary hypertension. 

### 4.5. Incident and Prevalent Patients

Most patients in the BNP-PL registry are prevalent cases which is due to the design of our study. As shown in Table 6, incident patients were older and suffered from more comorbid conditions as compared to prevalent ones. Additionally, CTD-PAH was more commonly diagnosed than CHD-PAH. These changes reflect the changing demographics of the PAH population observed in the Western populations.

### 4.6. Strengths and Limitations

Our study has several strengths. First of all, our study has enrolled a large group of patients in the era of modern PAH therapies over 10 years after major Europe and US based registries were started. Secondly, due to a specific reimbursement system, we present data for the whole population diagnosed and treated in Poland. Thirdly, we are the first to show the multicenter, prospectively collected data on PAH epidemiology in a Central-Eastern European country with a significantly different political and economic background than Western World countries. Previous reports from this geographical region presented retrospectively collected [40] or single center data [41]. 

Our study also has several limitations. The first is the small number of newly diagnosed as compared to previously diagnosed patients. Still, this proportion will change significantly as only incident cases will be enrolled in the follow-up part of our study. Secondly, as genetic testing has not been routinely performed in Polish patients, we have identified very few subjects with the BMPR2 gene mutation, which probably does not reflect the true prevalence of HPAH. Thirdly, the group of CTD-PAH patients seems to be underrepresented in our registry which probably results from the lack of national screening programmes in CTD population. Lastly, we merely collect data on the baseline characteristics of our patients rather than any analysis of their prognosis. This, however, will be available in the future follow-up analysis. The methodological limitations inherent to registry-based studies have been recently acknowledged [42].

## 5. Conclusions

Our study shows the epidemiology of PAH in a Central-Eastern post-communist European country with a relatively short history of availability of PAH therapies [32]. It estimates the prevalence of adult PAH at a similar level to that observed in the Western countries but highlights heterogeneity in the geographic distribution of PAH, which suggests that an important proportion of patients may still have not been diagnosed. Important differences between prevalent and incident cases show similar demographic trends as in the Western Europe registries. However, it is worth noting the high proportion of patients with CHD-PAH and the relatively low proportion of those with CTD-PAH. Double combination therapy is shown to be used as frequently as in registries from the Western European countries; nonetheless, the use of triple combination therapy even in high risk patients is relatively low. This may be more a reflection of the strict reimbursement conditions rather than any medical neglect, and therefore could be an issue for improvement within the reimbursement policy. This is counterbalanced by a relatively high frequency of parenteral prostanoid therapy in the most severe patients, higher than in previously published registries. 

## Figures and Tables

**Figure 1 jcm-09-00173-f001:**
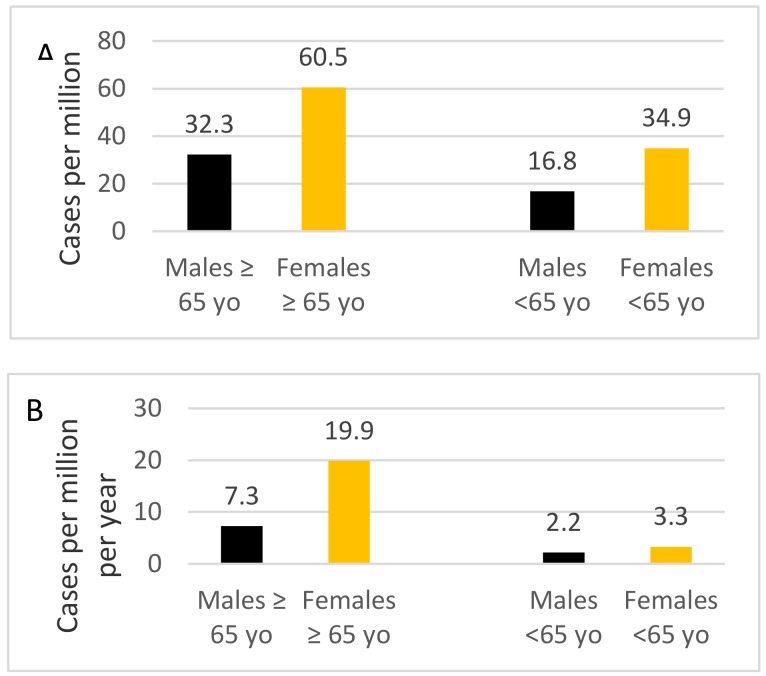
Prevalence (**A**) and incidence rate (**B**) of patients with PAH categorized by age and sex.

**Figure 2 jcm-09-00173-f002:**
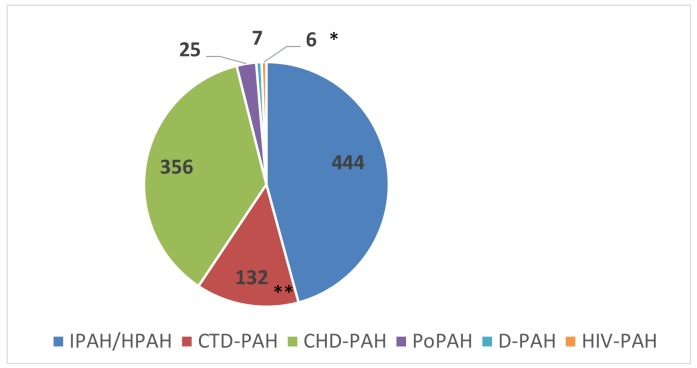
Number of patients with different types of pulmonary arterial hypertension according to clinical classification. In the group of IPAH/HPAH only 12 patients were diagnosed with HPAH based on genetic testing, however PAH patients were not systematically checked for genetic background. CHD-PAH–pulmonary arterial hypertension (PAH) associated with congenital heart disease, CTD-PAH-PAH associated with connective tissue disease, DPAH—drugs and toxin induced PAH, HIV-PAH—PAH associated with HIV infection, HPAH—heritable PAH, IPAH—idiopathic PAH, PoPAH—PAH associated with portal hypertension. * Among patients with CTD-PAH (*n* = 132), the most frequent were diseases of the scleroderma spectrum (SSc-PAH, *n* = 63) followed by mixed connective tissue disease (*n* = 22), rheumatoid arthritis (*n* = 15), systemic lupus erythematosus (*n* = 13), polimiositis (*n* = 5), and other conditions including overlap syndromes, Sjogren′s syndrome, undifferentiated systemic rheumatic disease, and dermatomyositis (*n* = 14). ** In 7 patients, PAH was attributed to the use of drugs and toxins including roxadustat (*n* = 1), dasytinib (*n* = 3), interferon (*n* = 2), and amphetamine (*n* = 1).

**Table 1 jcm-09-00173-t001:** Patients with Pulmonary Arterial Hypertension in the Data Base of Pulmonary Hypertension in the Polish Population (BNP-PL) Registry.

	All	I/HPAH	CTD	CHD	Portal	Drugs/ Toxins	HIV
No of patients	970	444	132	356	25	7	6
Age (years)	46.8 ± 22.3	54.6 ± 18.2	60.4 ± 13.9	31.5 ± 21.6	53.7 ± 21.6	57.1 ± 14.4	37.7 ± 5.0
Data acquired at diagnosis							
WHO FC, n (%)							
I	6 (0.6)	3 (0.8)	1 (0.8)	2 (0.6)	0	0	0
II	159 (16.4)	76 (17)	17 (12.9)	56 (15.7)	6 (24.0)	2 (28.6)	2 (33.3)
III	699 (72.1)	293 (66)	96 (72.7)	289 (81.2)	16 (64.0)	2 (28.6)	3 (50.0)
IV	106 (10.9)	72 (16.2)	18 (13.6)	9 (2.5)	3 (12.0)	3 (42.8)	1 (16.7)
Age (years)	53.8 ± 17.9	58.7 ± 16.9	63.8 ± 13.1	43.8 ± 16.3	56.4 ± 14.6	58.2 ± 12.9	41.1 ± 5.4
Data acquired at enrollment							
Female, n (%)	677 (69.8)	318 (71.6)	115 (87.1)	223 (62.4)	15 (60)	3 (42.9)	3 (50)
Incident cases, n (%)	96 (9.9)	48 (10.8)	20 (15.1)	19 (5.3)	5 (20)	1 (14.3)	3 (50)
WHO FC, n (%)							
I	49 (5.1)	28 (6.5)	6 (1.5)	13 (4.8)	0	2 (28.6)	0
II	455 (46.9)	200 (45.3)	52 (44.8)	187 (49.4)	12 (48)	2 (28.6)	2 (33.3)
III	429 (43.8)	195 (43.4)	64 (48.5)	151 (41.8)	13 (52)	2 (28.6)	4 (66.7)
IV	37 (4)	21 (3.8)	10 (5.2)	5 (4)	0	1 (14.3)	0
6 MWD [m]	375 ± 142	383 ± 151	344 ± 138	375 ± 132	376 ± 143	415.2 ± 74	408 ± 159
BMI [kg/m^2^]	25.8. ± 5.5	27.3 ± 5.4	25.3 ± 5.1	23.9 ± 5.4	26.1 ± 4.5	26.4 ± 7.9	22.6 ± 4.9
NTproBNP [ng/l] *	1446 ± 2733	1480 ± 2815	1881 ± 2526	1262 ± 2566	910 ± 1923	4328 ± 6926	2529 ± 2834
BNP [ng/l] *	310 ± 648	382 ± 761	316 ± 538	237.8 ± 558	-	-	-
mPAP [mmHg]	50.6 ± 18.9	45.1 ± 13.9	38.4 ± 12.6	63.4 ± 21.8	45.7 ± 8.8	41.0 ± 19.7	46.5 ± 13.8
PVR [WU]	7.8 ± 4.2	8.1 ± 4.2	7.0 ± 4.2	14.9 ± 9.9	6.8 ± 2.8	6.7 ± 5.6	8.8 ± 5.2
CI [l/min/m^2^]	2.7 ± 0.8	2.7 ± 1.8	2.7 ± 0.7	2.5 ± 1.1	3.2 ± 1.2	2.8 ± 1.0	2.7 ± 0.9
RAP [mmHg]	6.7 ± 4.3	6.7 ± 4.3	6.2 ± 4.3	7.0 ± 4.1	7.5 ± 4.3	8.2 ± 3.5	7.5 ± 5.3
PAWP [mmHg]	8.9 ± 3.3	9.4 ± 3.3	9.3 ± 3.2	8.2 ± 3.4	9.3 ± 3.5	7.0 ± 2.5	8.2 ± 3.5
RAA [cm^2^]	23.9 ± 9.2	24.6 ± 9.2	23.9 ± 9.8	22.4 ± 7.45	22.4 ± 8.7	23.9 ± 5.9	21.4 ± 8.6

* patients with chronic dialysis were excluded from the assessment of brain natriuretic peptide; in two centres BNP was used instead of NT-proBNP. BMI—body mass index, CI—cardiac index, mPAP—mean pulmonary artery pressure, 6 MWD—six min walking distance, NT-proBNP—N-terminal pro-B-type Natriuretic Peptide, PAWP—pulmonary artery wedge pressure, PVR—pulmonary vascular resistance, RAA—right atrial area, RAP – right atrial pressure, WHO FC—World Health Organization Functional Class WU—Wood unit.

**Table 2 jcm-09-00173-t002:** The prevalence of different heart defects in four subgroups of patients according to the European Society of Cardiology.

	Eisenmenger’s Syndrome *n* = 265 (75%)	PAH Associated with Prevalent Systemic-to-Pulmonary Shunts *n* = 43 (12%)	PAH after Defect Correction *n* = 45 (12%)	PAH with Small/Coincidental Defects *n*= 3 (1%)
ASD	39	18	21	1
VSD	121	8	8	1
AVSD	42	4	2	0
PDA	22	5	4	0
Complex defects	41	8	10	1

ASD—atrial septal defect, AVSD—atrioventricular septal defect, PAH—pulmonary arterial hypertension, PDA—persistent ductus arteriosus, VSD—ventricular septal defect.

**Table 3 jcm-09-00173-t003:** Proportions of Prevalent Patients with Risk Determinants of Pulmonary Arterial Hypertension Mortality in their Low, Intermediate and High Risk Range.

Risk Determinants	Low Risk	Intermediate Risk	High Risk
WHO FC	52%	44%	4%
6 MWD	41.7%	53%	5.3%
BNP/NT-proBNP	40.6%	30.8%	28.6%
RAA	25.7%	44.8%	29.5%
RAP	65.9%	28.7%	5.4%
CI	57.9%	22.7%	19.4%

The low, intermediate and high risk ranges for the prognostic parameters are as follows: WHO FC I and II, III, IV; 6 MWD >440 m, 165–440 m, <165 m; NT-proBNP <300 ng/L, 300–1400 ng/L, >1400 ng/L; BNP<50 ng/l50–300 ng/L, >300 ng/L; RAA < 18 cm^2^, 18–26 cm^2^, >26 cm^2^; RAP <8 mm Hg, 8–14 mm Hg, >14 mm Hg; CI >2.5 kg/min/m^2^, 2.0-2.5 kg/min/m^2^, <2.0 kg/min/m^2^. CI—cardiac index, WHO FC—World Health Organization Functional Class, NT-proBNP—N-terminal pro-B-type Natriuretic Peptide, 6 MWD—six min walking distance, RAA—right atrial area, RAP—right atrial pressure.

**Table 4 jcm-09-00173-t004:** PAH-Specific Medications Among Adult PAH Patients in the Data Base of Pulmonary Hypertension in the Polish Population (BNP-PL) Registry at Enrollment.

All Patients	Whole Sample:	I/HPAH	CTD	CHD	Portal	Drugs/Toxins	HIV
Monotherapy (*n*,%)	366 (37.7)	123 (27.7)	58 (43.9)	163 (45.8)	15 (60)	3 (42.9)	4 (66.7)
-Oral (*n*,%)	356 (36.7)	117 (26.3)	57 (43.2)	160 (44.9)	15 (60)	3 (42.9)	4 (66.7)
PDE5-I (*n*,%)	201	105	22	54	13	3	4
ERA (*n*,%) *	148	4	141	1	2	0	0
-Prostacyclin analogue (*n*,%)	10 (1)	6 (1.3)	1 (0.7)	3 (0.8)	0	0	0
Double comb. therapy (*n*,%)	515 (53)	254 (57.2)	72 (54.5)	173 (48.6)	10 (40)	4 (57.1)	2 (33.3)
-Oral (*n*,%) *	295 (30.4)	118 (26.6)	33 (27.3)	141 (39.6)	2 (8)	0	1 (16.7)
-With prostacyclin analogue (*n*,%)	220 (22.7)	136 (30.6)	39 (29.5)	32 (8.9)	8 (32)	4 (57.1)	1 (16.7)
Triple combination therapy (*n*,%)	46 (4.7)	29 (6.5)	8 (6)	9 (2.5)	0	0	0
Patients with WHO FC IV:							
Monotherapy (*n*,%)	7 (18.9)	3 (14.3)	0	4 (80)	0	0	0
-Oral (*n*,%)	7 (18.9)	3 (14.3)	0	4 (80)	0	0	0
-Prostacyclin analogue (*n*,%)	0	0	0	0	0	0	0
Double combination therapy (*n*,%)	21 (56.8)	13 (61.9)	7 (70)	0	0	1 (100)	0
-Oral (*n*,%)	5 (13.5)	3 (14.3)	2	0	0	0	0
-With prostacyclin analogue (*n*,%)	16 (43.2)	10 (47.6)	5	0	0	1 (100)	0
Triple combination therapy (*n*,%)	9 (24.3)	5 (23.8)	3 (30)	1 (20)	0	0	0

* Of 356 patients who used oral monotherapy 349 (98%) were receiving sildenafil or endothelin receptor antagonist. The other seven patients were receiving riociguat or selexipag which were available only in open label phases of clinical studies and were not formally reimbursed by the National Health Fund. ** Of 295 patients who used oral double combination therapy 292 (99%) patients used endothelin receptor antagonist and sildenafil, three other patients were using a combination of riociguat and bosentan, or selexipag and sildenafil. ERA—endothelin receptor antagonist, PDE5-I—phosphodiesterase type 5 inhibitor, WHO FC—World Health Organization Functional Class.

**Table 5 jcm-09-00173-t005:** Concomitant diseases and their treatments.

	All Patients	I/HPAH	CTD	CHD	Portal	Drugs/Toxins	HIV
Hypertension (*n*,%)	369 (38.0)	222 (50.0)	71 (53.8)	62 (17.4)	10 (40)	2 (28.6)	2 (33.3)
Underweight (*n*,%)	53 (5.5)	7 (1.6)	9 (6.8)	36 (10.1)	0	0	1 (16.7)
Overweight (*n*,%)	300 (30.9)	147 (33.1)	46 (34.8)	94 (26.4)	12 (0.48)	1 (14.2)	0
Obesity (*n*,%)	197 (20.3)	130 (36.5)	20 ((15,1)	40 (11.2)	4 (0.16)	2 (28.4)	1 (16.7)
Diabetes mellitus (*n*,%)	154 (15.8)	109 (24.5)	27 (20.5)	13 (3.6)	3 (12)	1 (14.3)	1 (16.7)
Smoking							
-Active (*n*,%)	41 (4.2)	23 (5.2)	2 (1.5)	10 (2.8)	3 (12)	1 (14.3)	2 (33.3)
-Previously (*n*,%)	192 (19.8)	116 (26.1)	27 (20.5)	35 (9.8)	8 (0.32)	2 (28.6)	4 (66.6)
CAD (*n*,%)	113 (11.6)	82 (18.5)	17 (12.9)	10 (2.8)	2 (8)	1 (14.3)	1 (16.7)
Myocardial infarction (*n*,%)	42 (4.3)	28 (6.3)	9 (6.8)	3 (0.8)	0	1 (14.3)	1 (16.7)
Depression (*n*,%)	55 (5.6)	21 (4.7)	14 (10.6)	7 (4.8)	3 (12)	0	0
COPD (*n*,%)	73 (7.5)	45 (10.1)	8 (6.1)	20 (5.6)	0	0	0
Asthma (*n*,%)	54 (5.6)	27 (6)	4 (3)	22 (6.2)	0	1 (14.3)	0
Hypothyroidism (*n*,%)	212 (21.9)	76 (17.1)	31 (23.4)	101 (28.3)	3 (12)	1 (14.3)	0
Hyperthyoidism (*n*,%)	26 (2.7)	18 (4.1)	2 (1.5)	6 (1.7)	0	0	0
Liver cirrhosis (*n*,%)	32 (3.3)	9 (2.0)	0	1 (0.3)	20 (80)	0	2 (33.3)
CKD (n,%)	147 (15.2)	82 (18.5)	27 (20.5)	34 (9.6)	0	1 (14.3)	3 (50)
Dialysis (*n*,%)	5 (0.5)	4 (0.9)	0	0	0	0	1 (16.7)
Endocarditis history (*n*,%)	9 (0.9)	2 (0.5)	1 (0.8)	6 (1.7)	0	0	0
Atrial fibrillation (*n*,%)	166 (17.1)	87 (19.6)	18 (13.6)	56 (15.7)	3 (12)	2 (28.6)	0
Atrial flutter (*n*,%)	52 (5.4)	25 (5.6)	5 (3.8)	22 (6.2)	0	0	0
Down syndrome (*n*,%)	95 (9.8)	2 (0.5)	1 (0.8)	92 (25.8)	0	0	0
Mental retardation (*n*,%)	106 (10.9)	5 (1.1)	1 (0.8)	100 (28.1)	0	0	0
Cancer history (*n*,%)	63 (6.5)	39 (8.9)	10 (7.6)	9 (2.5)	3 (12)	2 (28.6)	0
PE history (*n*,%)	35 (3.6)	22 (4.9)	5 (3.8)	6 (1.7)	1 (4)	0	1 (16.7)
Sleep apnea syndrome (*n*,%)	14 (1.4)	11 (2.5)	1 (0.8)	1 (0.3)	0	1 (14.3)	0
Cardiac pacing (*n*,%)	31 (3.2)	13 (2.9)	3 (2.2)	15 (4.2)	0	0	0
Therapies (*n*,%):							
Home oxygen therapy	103 (10.6)	60 (13.5)	26 (19.7)	16 (4.5)	1 (4)	0	0
Vitamin K antagonists	181 (18.7)	98 (22.1)	15 (11.4)	66 (18.5)	0	2 (28.6)	0
Low-molecular heparin	27 (2.8)	12 (2.7)	4 (3)	9 (2.5)	1 (4)	0	1 (16.7)
New oral anticoagulants	86 (8.9)	47 (10.6)	16 (12.1)	22 (6.2)	1 (4)	0	0
Beta blockers *	396 (40.8)	170 (38.3)	66 (50.0)	137 (38.4)	18 (72)	2 (28.6)	3 (50)
ACEI	186 (19.2)	96 (21.6)	34 (25.8)	47 (13.2)	5 (20)	2 (28.6)	2 (33.3)
ARB	67 (6.9)	36 (8.6)	16 (12.1)	15 (4.2)	0	0	0
Ivabradine	20 (2.1)	13 (2.9)	2 (1.5)	4 (1.1)	0	1 (14.3)	0
Amiodarone	32 (3.3)	20 (4.5)	2 (1.5)	9 (2.5)	1 (4)	0	0
Other antiarrhythmics	68 (7)	32 (7.2)	10 (7.6)	26 (7.3)	0	0	0
Loop diuretics	383 (39.5)	130 (29.3)	43 (35.6)	200 (56.2)	6 (24)	2 (28.6)	2 (33.3)
Thiazide diuretics	75 (7.7)	42 (9.5)	13 (9.8)	18 (5.1)	2 (8)	0	0
Potassium-sparing diuretics	368 (37.9)	173 (38.5)	49 (37.1)	124 (34.8)	15 (60)	4 (57.1)	3 (50)
SSRI	47 (4.8)	19 (4.3)	6 (4.5)	19 (5.3)	2 (8)	1 (14.3)	0
Other antidepressants	38 (3.9)	20 (4.5)	11 (8.3)	7 (1.9)	0	0	0
ASA	140 (14.4)	71 (15.9)	25 (18.9)	40 (11.2)	2 (8)	1 (14.3)	1 (16.7)
Clopidogrel	15 (1.5)	10 (2.3)	2 (1.5)	2 (0.6)	0	1 (14.3)	0
Proton pomp inhibitors	366 (38.9)	190 (42.7)	89 (67.4)	66 (18.5)	15 (60)	3 (42.9)	3 (50)
Statins	270 (27.8)	174 (39.1)	39 (29.5)	54 (15.2)	2 (8)	1 (14.3)	0
Corticosteroids	87 (8.9)	19 (4.5)	59 (44.7)	4 (1.1)	4 (16)	0	1 (16.7)
Immunosuppressive drugs	68 (7.0)	10 (2.2)	49 (37.1)	2 (0.6)	3 (12)	2 (28.6)	2 (33.3)

ACEI—angiotensin convertase inhibitors, ARB—angiotensin receptor blockers, ASA—acetylsalicylic acid, CAD—coronary artery disease, CHD—congenital heart disease, CKD—chronic kidney disease, COPD—chronic obstructive pulmonary disease, CTD—connective tissue disease, IPAH—idiopathic pulmonary arterial hypertension, HPAH—hereditary pulmonary arterial hypertension, PE—pulmonary embolism, SSRI—Selective Serotonin Reuptake Inhibitors. * The following beta-blockers were used by PAH patients in the present study: bisoprolol (*n* = 165, 17%), nebivolol (*n* = 92; 9.5%), metoprolol (*n* = 90; 9.3%), carvedilol (*n* = 30; 3.1%), propranolol (*n* = 10; 1%), sotalol (*n* = 5; 0.5%), betaxolol (*n* = 4; 0.4%).

**Table 6 jcm-09-00173-t006:** Comparison of incident and prevalent patients with pulmonary arterial hypertension.

	Incident Cases	Prevalent Cases	*p*
Number	96	874	
Age [years]	62.7±16.9	52.7±17.7	<0.0001
Female sex (*n*,%)	69(71.8)	608(69.6)	0.6
BMI [kg/m^2^]	27.4±5.6	25.7±5.4	0.0003
Diagnosis (*n*,%):			
IPAH/HPAH	48 (50.0)	396 (45.3)	0.03
CTD-APAH	20(20.8)	112(12.8)	0.03
CHD-APAH	19(19.8)	337(38.8)	0.0003
WHO-FC			
I	1(1)	48(5.5)	<0.0001
II	15(15.6)	440(50.3)	
III	63(65.6)	366(42.9)	
IV	17(17.7)	20(2.3)	
6MWD [m]	289 ± 136	384 ± 140	<0.0001
NTproBNP [ng/L]	3133.7 ± 5367.9	1670.1 ± 7653.8	0.07
RAA [cm^2^]	25.3 ± 8.1	23.8 ± 9.3	0.16
Right heart catheterization:			
CI [l/min/m^2^]	2.4 ± 0.95	2.8 ± 0.86	0.22
mPAP [mmHg]	45.5 ± 15.1	51.2 ± 19.3	0.006
PVR [WU]	8.9 ± 4.5	9.7 ± 7.3	0.35
RAP [mmHg]	7.4 ± 4.7	6.7 ± 4.4	0.61
Concomitant diseases (n,%):			
Hypertension	55(57.3)	314(35.9)	<0.0001
Diabetes mellitus	28(29.1)	126(14.4)	0.002
Smoking	6(6.3)	35(4.0)	0.3
Previous smoking	33 (34)	159 (18)	0.0002
Coronary artery disease	21(21.8)	92(10.5)	0.001
Myocardial infarction	9(9.4)	33(3.8)	0.01
Chronic kidney disease	19(19.8)	128(14.6)	0.18
Atrial fibrillation	24(25.0)	142(16.2)	0.08
Atrial flutter	4(4.2)	48(5.5)	0.88

BMI—body mass index, IPAH—idiopathic pulmonary arterial hypertension, CHD—congenital heart disease, CI—cardiac index, CTD—connective tissue disease, mPAP—mean pulmonary artery pressure, 6 MWD—six min walking distance, NT-proBNP—N-terminal pro-B-type Natriuretic Peptide, PVR—pulmonary vascular resistance, RAA—right atrial area, RAP—right atrial pressure, WHO-FC—World Health Organization Functional Class.

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
