# Peer review of "Characterization of Patients with Pulmonary Arterial Hypertension: Data from the Polish Registry of Pulmonary Hypertension (BNP-PL)"

_jcm, 2020, doi:10.3390/jcm9010173_

Round 1

Reviewer 1 Report

In this original research manuscript, Kopeć and colleagues show data from the Polish registry of pulmonary hypertension and compare their observations with characteristics and demographics found in the western countries. Insights in PAH epidemiology of a central-eastern European country may be of great interest for the readership of J Clin Med. In this study, interesting findings are presented. However, important methodical information is lacking, and presentation and interpretation of the findings need to be improved.

Major Comments

Genetic testing is missing in most patients. Thus, the HPAH patients are most probably underrepresented. Thus, underrepresentation of HPAH patients as well as the probable underrepresentation of CTD-PAH patients should be mentioned in the introduction or in the “experimental section” of this manuscript to allow a better interpretation of the data. In Figure 2, for adequate interpretation, the number of HPAH patients needs to be shown separately from the IPAH patients. In Figure 2, a combined IPAH/HPAH group should be avoided. The M&M section lacks very important information, which makes it impossible for the reader to adequately interpret the results. The authors need to state how exactly the IPAH patients were classified. Has e.g. (high-resolution) CT of the chest been performed in all of the patients? Which analyses were performed to rule out CTEPH? Were those analyses performed in all patients classified as IPAH? Were patients classified as IPAH systematically screened for CTD? The authors need to state how exactly the HPAH patients were classified. By screening for BMPR2 mutations only? Any other gene tests beside BMPR2? By sole medical history in some patients? Which analyses were performed to distinguish between severe PH-COPD and IPAH, since 10% of H/IPAH patients suffered from COPD? In line 165, please specify “other conditions” and name those conditions. In lines 291-293 the authors critically discuss possible reasons for the relatively low number of CTD-PAH patients. In addition, the probable underrepresentation of the CTD-PAH group should also be discussed in the “strengths and limitations” section.

Minor Comments

Please compare your results on home oxygen therapy in PAH patients (10.6%) with published data in other countries. Which t test was used in this study? Did you analyze for normal distribution? If so, which test was used? In Table 1, the percentages are partially incorrect (“WHO FC I-IV Portal” and “WHO FC I-IV HIV”). Please correct. In Table 1, please correct “l/mn/m²“ to “l/min/m²”. In Table 6, it should probably write “I/HPAH” instead of “IPAH”.

Author Response

Reviewer 1.

Reviewer’s comment 1

Genetic testing is missing in most patients. Thus, the HPAH patients are most probably underrepresented. Thus, underrepresentation of HPAH patients as well as the probable underrepresentation of CTD-PAH patients should be mentioned in the introduction or in the “experimental section” of this manuscript to allow a better interpretation of the data.

In Figure 2, for adequate interpretation, the number of HPAH patients needs to be shown separately from the IPAH patients. In Figure 2, a combined IPAH/HPAH group should be avoided.

Authors’ response 1

As genetic testing are not routinely performed in PAH patients in Poland we cannot currently conclude from our registry about the prevalence of heritable PAH (HPAH). Accordingly, IPAH patients and patients in whom HPAH was diagnosed have been considered as a single group. According to Reviewer’s note we added this information in the “Experimental section” of our manuscript.

We additionally  added the following information in the footnote of Figure 2 “In the group of IPAH/HPAH only 12 patients were diagnosed with IPAH based on genetic testing, however PAH patients were not systematically checked for genetic background.”

Reviewer’s comment 2

The M&M section lacks very important information, which makes it impossible for the reader to adequately interpret the results. The authors need to state how exactly the IPAH patients were classified. Has e.g. (high-resolution) CT of the chest been performed in all of the patients? Which analyses were performed to rule out CTEPH? Were those analyses performed in all patients classified as IPAH? Were patients classified as IPAH systematically screened for CTD? The authors need to state how exactly the HPAH patients were classified. By screening for BMPR2 mutations only? Any other gene tests beside BMPR2? By sole medical history in some patients?

Authors’ response 2.

The main goal  of our registry is  to  describe  current practice and outcomes in patients with PAH therefore the registry is entirely observational. All data is acquired from patient records available at the treating centers. Importantly, the protocol of the study does not require any additional patients’ visits or diagnostic tests, and does not influence the management of patients.  We also did not prepare any specific protocol for genetic testing for the purpose of the registry, however a common protocol for genetical testing has been established in some centers as a part of another project which has been recently opened.

In our registry, the diagnosis of IPAH was at the discretion of coinvestigators who managed the enrolled  patients in their centres. However, the diagnosis must have fulfilled the criteria recommended by the European Society of Cardiology. This means that patients were diagnosed for PAH when left heart disease, chronic thromboembolic disease and pulmonary diseases were excluded. IPAH could have been diagnosed only in patients in whom congenital heart defect, connective tissue disease, HIV infection, portal hypertension and possible relation to drugs were excluded.

We enrolled in our study reference centers of PAH accredited by the National Health Fund which ensures validity and credibility of the diagnostic algorithms. However, we can not exclude some variability in the diagnostic approach of different centres which we did not study in detail for the purpose of the present analysis.

The protocol of our registry has been described previously in Kardiol Pol. (2019; 77: 972-974) and in the Appendix A to the present manuscript therefore we did not describe it extensively in the main text of the present study. However according to Reviewer’s note we supplemented the Experimental section of the present manuscript.

Reviewer’s comment 3

Which analyses were performed to distinguish between severe PH-COPD and IPAH, since 10% of H/IPAH patients suffered from COPD?

Authors’ response 3

In our registry we found that a significant number of patients (n=127, 13%) had a diagnosis of COPD or asthma.  Similar proportions (21.9%) were shown in the largest PAH registry conducted in the United States (REVEAL registry). This means that despite suffering from obstructive lung diseases some patients had been diagnosed with group 1 pulmonary hypertension (PAH). In that way COPD or asthma in a mild form were considered as a comorbid condition in a PAH patient rather than as etiology of pulmonary hypertension. This approach has been recently supported by the Experts of the 6th World Symposium on Pulmonary Hypertension. They suggest that in patients who have pulmonary hypertension and obstructive lung disease with FEV1>60% and minimal parenchymal changes the primary diagnosis should be PAH rather than group 3 pulmonary hypertension.

Final decision on the diagnosis of individual patient was however at the discretion of the treating physicians and as stated above we did not influence the protocols used in different centers which would make our study more interventional than observational. The methodological limitations  inherent to registry-based studies have been recently acknowledged (Weatherald J, Reis A, Sitbon O, Humbert M. Pulmonary arterial hypertension registries: past, present and into the future. Eur Respir Rev. 2019;28(154):190128).

Reviewer’s comment 4

In line 165, please specify “other conditions” and name those conditions.

Authors’ response 4

This group included patients with overlap syndromes, Sjogren’s syndrome, undifferentiated systemic rheumatic disease, and dermatomyositis.

We added this information in the footnote below Figure 2.

Reviewer’s comment 5

In lines 291-293 the authors critically discuss possible reasons for the relatively low number of CTD-PAH patients. In addition, the probable underrepresentation of the CTD-PAH group should also be discussed in the “strengths and limitations” section.

Authors’ response 5

We added the following sentence to the limitations section of our manuscript: “the group of CTD-PAH patients seems to be underrepresented in our registry which probably results from the lack of national screening programmes in CTD population”

Reviewer’s comment 6

Please compare your results on home oxygen therapy in PAH patients (10.6%) with published data in other countries.

Authors’ response 6

According to Reviewer’s note we compared the use of oxygen supplementation in our study and in other PAH registries in the Discussion section of the manuscript as follows:

“We found that the proportion of PAH patients who were using home oxygen therapy in our study was 10.6% which is low when compared to data from other registries including the US REVAL registry (40.3%), the SPAHR registry (33%) and the Spanish registry of patients treated with inhaled iloprost (38%). This may result from several factors including the lack of data from randomized trials to support oxygen supplementation in PAH subjects, quite liberal recommendations of the ESC for oxygen use in PAH (use only in patients with blood O2 pressure consistently > 60 mmHg), and a high proportion of patients with CHD-PAH in our study. According to the ESC guidelines in the latter subpopulation of PAH patients the use of supplemental O2 therapy should be considered only in cases in which it produces a consistent increase in arterial O2 saturation and reduces symptoms. We can not also exclude the role of some difficulties with access to oxygen generators”

Reviewer’s comment 7

Which t test was used in this study? Did you analyze for normal distribution? If so, which test was used?

Authors’ response 7

We used statistical analysis for the comparison of incident and prevalent cases. According to the Reviewer’s note we completed the Statistical analysis part of our manuscript with the following information: “Continuous  variables were checked for normal distribution with chi-square test. For the comparison of continuous variables between 2 groups, we used the Student’s t test”.

Reviewer’s comment 8

In Table 1, the percentages are partially incorrect (“WHO FC I-IV Portal” and “WHO FC I-IV HIV”). Please correct. In Table 1, please correct “l/mn/m²“ to “l/min/m²”. In Table 6, it should probably write “I/HPAH” instead of “IPAH”.

Authors’ response 8

We are very sorry for these mistakes. We corrected them according to Reviewer’s suggestion.

Reviewer 2 Report

Comment to the authors jcm-672831

Grzegorz Kopeć and colleagues characterized the patients with pulmonary arterial hypertension (PAH) from the Polish registry of pulmonary hypertension (BNP-PL). Unlike the majority of the study performed on the western world-based population, this study characterizes a group of Caucasian PAH adults of the central Eastern European country. They analyzed the data from the multicenter prospective Polish registry and described the demographics, treatment and the burden of coexisting diseases of the PAH patient involved in the BNP-PL. The reported a similar prevalence, demographic than the one previously described for the western countries. However, they observed a high proportion of patients with chronic heart disease PAH (CHD-PAH) and a low proportion of patients with connective tissue disease PAH (CTD-PAH). Unlike the western countries, they also reported a low proportion of triple therapy treated patients in the BNP-PL cohort. Whilst this study exclusively descriptive and described the baseline characteristic of the BNP-PL, it is an elegant, interesting and well-written work. I only have minor comments.

In the discussion section (line 269, 270), the authors mentioned that “Polish males live 7.8 years shorter than Polish females”. It would be useful to add the data relative to the mortality/survival in the manuscript.

Compared to the western cohorts, the use of triple therapy in the BNP-PL cohort is lower. It would interesting to know/discuss how this difference impacts the overall survival of the BNP-PL cohort.

The authors addressed the % of patients treated with prostacyclin analogs (Table 4). In the same table, It would be interesting the proportion of patients treated with the other PAH therapies (e.g. Calcium channel blocker, endothelin receptor antagonist, PDE5 inhibitors, guanylate cyclase stimulator).

The authors reported a higher proportion of patients treated with beta-blockers in the BNP-PL cohort. As the authors are certainly aware there are some concerns regarding the benefit/deleteriousness of beta-blocker in PAH patients (Perros et al. Circulation : Heart Failure; 2017). The authors should consider reporting the class of beta-blockers used (Non-selective, beta 1-selective, with alpha-blocking activity). Besides, it would be interesting to discuss the impact of beta-blockers on the outcome of the patients (e.g. what is the mPAP, 6MWD, mortality in treated and nontreated PH patients)

5% of BNP-PL patients are diagnosed with COPD. Why these patients categorized as group-1 PH rather than group-3 PH ?

Minor typos should be corrected. E.g. line 354 reference “6” is not properly formatted.

Author Response

Reviewer 2.

Reviewer’s note 1

In the discussion section (line 269, 270), the authors mentioned that “Polish males live 7.8 years shorter than Polish females”. It would be useful to add the data relative to the mortality/survival in the manuscript.

Authors’ response 1

In the present study we present characteristic of our PAH patients. Our registry started in March 2018 and is a prospective study therefore currently we do not have mortality/survival data. As we plan to follow our group for 10 years (Kardiol Pol. 2019; 77: 972-974) we will be able to analyze the role of several parameters on survival of PAH patients in the future. In such analysis we will be able to relate survival in PAH to the overall survival in Polish population.

Reviewer’s note 2

Compared to the western cohorts, the use of triple therapy in the BNP-PL cohort is lower. It would interesting to know/discuss how this difference impacts the overall survival of the BNP-PL cohort.

Author’s response 2

In Poland the access to PAH specific therapies is regulated by the National Health Fund. At the time of enrollment to our study triple combination therapy was not reimbursed for PAH patients, therefore only a very small number of patients were treated with a combination of PDE5i, ERA and prostacyclin analogues. Triple combination has started to be reimbursed in the late 2018. According to Reviewer’s suggestion we plan to analyze different reimbursement policies and the availability of triple combination on the overall survival of the BNP-PL cohort. This however will require long term observation of our patient which we plan for 10 years as previously presented (Kardiol Pol. 2019; 77: 972-974).

Reviewer’s note 3

The authors addressed the % of patients treated with prostacyclin analogs (Table 4). In the same table, It would be interesting the proportion of patients treated with the other PAH therapies (e.g. Calcium channel blocker, endothelin receptor antagonist, PDE5 inhibitors, guanylate cyclase stimulator).

Authors’ response 3

According to Reviewer’s suggestion we completed Table 4 with the information about different classes of PAH targeted therapies used in oral monotherapy and oral double combination therapy. Of 356 patients who used oral monotherapy 349 (98%) used sildenafil or endothelin receptor antagonist. The other 7 patients were receiving riociguat or selexipag which were available only in open label phases of clinical studies and were not formally reimbursed by the National Health Fund. Of 295 patients who used oral double combination therapy 292 (99%) patients used endothelin receptor antagonist and sildenafil, 3 other patient were using a combination of riociguat and bosentan  or selexipag and  sildenafil.

In part 3.5 of the manuscript we showed that 39 patients with reactive IPAH received calcium channel blockers to treat PAH.

Reviewer’s note 4

The authors reported a higher proportion of patients treated with beta-blockers in the BNP-PL cohort. As the authors are certainly aware there are some concerns regarding the benefit/deleteriousness of beta-blocker in PAH patients (Perros et al. Circulation : Heart Failure; 2017). The authors should consider reporting the class of beta-blockers used (Non-selective, beta 1-selective, with alpha-blocking activity). Besides, it would be interesting to discuss the impact of beta-blockers on the outcome of the patients (e.g. what is the mPAP, 6MWD, mortality in treated and nontreated PH patients)

Authors’ response 4

In the BNP-PL registry a significant number of PAH patients (n=396; 40.8%) were treated with beta-blockers. This number corresponds with a relatively high occurrence of medical conditions which may require the use of beta-blockers including: coronary artery disease (11.6%), atrial fibrillation and flutter (22.5%), or hypertension (38%). Although beta-blockers are generally not advised in patients with PAH their use when required by co-morbidities (i.e. high blood pressure, coronary artery disease or left heart failure) is supported by the recent guidelines of the European Society of Cardiology. In our registry most PAH patients were receiving bisoprolol (n=165, 17%) which was followed by  nebivolol (n=92; 9.5%), metoprolol (n=90; 9.3%), carvedilol (n=30; 3.1%), propranolol (n=10; 1%), sotalol (n=5; 0.5%), betaxolol (n=4; 0.4%). According to the Reviewer’s suggestion we added the information about the use of different beta-blockers under Table 5 of our manuscript. Additionally, we discussed current knowledge and recommendations about the use of beta-blockers in PAH in Therapy, comorbid conditions and risk part of the Discussion section in our manuscript. We thank the Reviewer for the suggestion to asses the impact of different classes of beta-blockers on the outcome in PAH.  We will analyze this important point in the follow-up phase of our project.

Reviewer’s note 5

5% of BNP-PL patients are diagnosed with COPD. Why these patients categorized as group-1 PH rather than group-3 PH ?

Authors’ response 5

In our registry we found that a significant number of patients (n-127, 13%) had a diagnosis of COPD or asthma.  Similar proportions (21.9%) were shown in the largest PAH registry conducted in the United States (REVEAL registry). This means that despite suffering from obstructive lung diseases some patients had been diagnosed with group 1 pulmonary hypertension (PAH).  In that way COPD or asthma in a mild form were considered a comorbid condition in a PAH patient rather than etiology of pulmonary hypertension. This approach has been recently supported by the Experts of the 6th World Symposium on Pulmonary Hypertension. They suggest that in patients who have pulmonary hypertension and obstructive lung disease with FEV1>60% and minimal parenchymal changes the primary diagnosis should be PAH rather than group 3 pulmonary hypertension.

Reviewer 3 Report

Kopeć and collages provide a comprehensive manuscript detailing the clinical and therapeutic characteristics patients with PAH in the Polish registry of pulmonary hypertension.

The major strengths are the large numbers of patients, prospective collection of high quality data from specialist PH centres that represent a full National cohort. The data includes excellent information on patient risk, medical therapy, co-morbidities and concomitant medication.

The demographics of patients in this cohort are in line with other large series and the recent changes identified are by enlarge represented here.

The under representation of patients with connective tissue disease associated PH is of interest. Is there a national screening program for patients with connective tissue disease in Poland - a number of countries recommend regular echo for early (ish) detection of PAH in this population.

Minor points that may improve readability / data representation:

6MWD in table 1 - the authors could reduce number of significant figures as the distance is likely only be measured to the nearest meter or half meter.

Variability in geographic data is also of great interest and has been identified as an issue in other countries with large specialist centres. This may be attributable to referral bias - might these data be better represented on a map with PH centres also indicated so distance from centre is represented (or some other means to provide the same data).

Author Response

Reviewer 3

Reviewer’s note 1

The under representation of patients with connective tissue disease associated PH is of interest. Is there a national screening program for patients with connective tissue disease in Poland - a number of countries recommend regular echo for early (ish) detection of PAH in this population.

Authors’ response 1

In our study the prevalence of pulmonary arterial hypertension associated with connective tissue disease was low when compared to the US and Western European registries. We suppose that it may result from a lack of an organized screening program for this group of patients. In Poland, patients with scleroderma spectrum disorders are  treated by rheumatologists or dermatologists. In most academic centres the cooperation between rheumatologists or dermatologists and PH specialists allows effective screening of these patients which include routine at least echocardiographic examination every year. However, the awareness of PH as a complication of scleroderma and the need for screening for PH may not be sufficient in non academic centres.

We added the following sentence to the Limitations section of the manuscript: “Thirdly, the group of CTD-PAH patients seems to be underrepresented in our registry which probably results from the lack of national screening programmes in CTD population.”

Reviewer’s note 2

6MWD in table 1 - the authors could reduce number of significant figures as the distance is likely only be measured to the nearest meter or half meter.

Authors’ response 2

According to the Reviewer’s note we  deleted the decimal places when expressing the 6 MWD.

Reviewer’s note 3                                                                                                

Variability in geographic data is also of great interest and has been identified as an issue in other countries with large specialist centres. This may be attributable to referral bias - might these data be better represented on a map with PH centres also indicated so distance from centre is represented (or some other means to provide the same data).

Authors’ response 3

We share the Reviewer’s assumption that variability in geographic distribution of the prevalence of PAH is attributable to differences in patients’ referral to the PH centres. This can result from some variability in awareness of pulmonary hypertension among physicians from different parts of Poland. According to Reviewer’s suggestion we prepared a map to show the prevalence of PAH in different parts of Poland together with the number of PH reference centres localized in these regions. We also calculated a correlation between the number of centres in a region and the prevalence of PAH, however we did not find statistically significant results (r=0.28; p=0.29).
